# Identification and In-Depth Analysis of the Novel FGFR2-NDC80 Fusion in a Cholangiocarcinoma Patient: Implication for Therapy

**Alexander Scheiter [1,\*], Felix Keil [1], Florian Lüke [2,3], Jirka Grosse [4], Niklas Verloh [5], Sabine Opitz [6], Sophie Schlosser [7], Arne Kandulski [7], Tobias Pukrop [2], Wolfgang Dietmaier [1], Matthias Evert [1], Diego F. Calvisi [1] and Kirsten Utpatel [1]**

1.  Institute of Pathology, University of Regensburg, 93053 Regensburg, Germany; felix1.keil@klinik.uni-regensburg.de (F.K.); wolfgang.dietmaier@klinik.uni-regensburg.de (W.D.); matthias.evert@ukr.de (M.E.); diego.calvisi@klinik.uni-regensburg.de (D.F.C.); kirsten.utpatel@klinik.uni-regensburg.de (K.U.)
2.  Department of Internal Medicine III, Hematology and Oncology, University Hospital Regensburg, 93053 Regensburg, Germany; florian.lueke@klinik.uni-regensburg.de (F.L.); tobias.pukrop@klinik.uni-regensburg.de (T.P.)
3.  Fraunhofer-Institut für Toxikologie und Experimentelle Medizin ITEM-R, 93053 Regensburg, Germany
4.  Department of Nuclear Medicine, University Hospital Regensburg, 93053 Regensburg, Germany; jirka.grosse@klinik.uni-regensburg.de
5.  Department of Radiology, University Hospital Regensburg, 93053 Regensburg, Germany; niklas.verloh@ukr.de
6.  Department of Surgery, University Hospital Regensburg, 93053 Regensburg, Germany; sabine.opitz@klinik.uni-regensburg.de
7.  Department of Internal Medicine I, University Hospital Regensburg, 93053 Regensburg, Germany; sophie.schlosser@klinik.uni-regensburg.de (S.S.); arne.kandulski@klinik.uni-regensburg.de (A.K.)
\*   Correspondence: alexander.scheiter@klinik.uni-regensburg.de; Tel.: +49-941-944-6707

**Abstract:** Fibroblast growth factor receptor 2 (FGFR2) fusions have emerged as a new therapeutic target for cholangiocarcinoma in clinical practice following the United States Food and Drug Administration (FDA) approval of Pemigatinib in May 2020. FGFR2 fusions can result in a ligand-independent constitutive activation of FGFR2 signaling with a downstream activation of multiple pathways, including the mitogen-activated protein (MAPK) cascade. Until today, only a limited number of fusion partners have been reported, of which the most prevalent is BicC Family RNA Binding Protein (BICC1), representing one-third of all detected FGFR2 fusions. Nonetheless, in the majority of cases rare or yet unreported fusion partners are discovered in next-generation sequencing panels, which confronts clinicians with a challenging decision: Should a therapy be based on these variants or should the course of treatment follow the (limited) standard regime? Here, we present the case of a metastasized intrahepatic cholangiocarcinoma harboring a novel FGFR2-NDC80 fusion, which was discussed in our molecular tumor board. The protein NDC80 kinetochore complex component (NDC80) is an integral part of the outer kinetochore, which is involved in microtubule binding and spindle assembly. For additional therapeutic guidance, an immunohistochemical analysis of the predicted fusion and downstream effector proteins was performed and compared to cholangiocarcinoma samples of a tissue microarray. The FGFR2-NDC80 fusion resulted in strong activation of the FGFR2 signaling pathway. These supporting results led to a treatment recommendation of Pemigatinib. Unfortunately, the patient passed away before the commencement of therapy.

**Keywords:** cholangiocarcinoma; FGFR fusion; NDC80; FRS2

## 1. Introduction

Cholangiocarcinoma (CCA) is the second most prevalent primary liver cancer [1], and, except for surgery in the early stages, the treatment options remain scarce. The first

line systemic therapy consists of the combination of gemcitabine and cisplatin [2], while there is no established subsequent standard of care. The clinical benefit of second-line chemotherapy is modest [3]. Therefore, the approval of Pemigatinib in May 2020 by the FDA based on the results of the FIGHT-202 study for CCAs harboring a Fibroblast growth factor receptor 2 (FGFR2) fusion or rearrangement [4] represents a significant milestone both in the treatment of this cancer entity as well as in the field of personalized medicine. Fibroblast growth factor receptor (FGFR) fusions can be found in various cancers, including rearrangements of FGFR2 in 14 percent of CCAs. In comparison, Fibroblast growth factor receptor 3 fusions occur in 3 to 6 percent of bladder cancers, 3 percent of glioblastomas, 0.5 percent of lung adenocarcinomas, and 3 percent of squamous cell carcinomas of the lung [5]. In contrast, Fibroblast growth factor receptor 1 is mainly amplified with only rare translocation events reported in glioblastomas, breast cancer and lung squamous cell carcinomas [5]. Several more frequent fusion partners have been described, including BicC Family RNA Binding Protein 1 (BICC1), transforming acidic coiled-coil-containing protein (TACC1), transforming acidic coiled-coil-containing protein *3* (TACC3), BAR/IMD Domain Containing Adaptor Protein 2 Like 1 (BAIAP2L1), and Adenosylhomocysteinase Like 1 (AHCYL) [5], while the majority of detectable fusions are found individually and are yet unreported [4]. This can pose significant difficulties in the clinical evaluation of their pathogenicity, such as the case presented here. FGFR fusions are considered to develop oncogenic potential through different mechanisms, including the constitutive activation of the FGFR kinase domain and the mistargeting of the fusion protein to the spindle apparatus leading to genomic instability and increased expression due to transcriptional dysregulation [5]. The downstream effectors of FGFR include PI3K-AKT-mTOR as well as the MAPK-signaling pathway. Fibroblast growth factor receptor substrate 2 (FRS2) in a complex with growth factor receptor-bound protein 2 (GRB2) mediates the activation of these pathways. In parallel phosphoinositide phospholipase C (PLC) is also activated [5]. Interestingly, NDC80 kinetochore complex component (NDC80), the fusion partner to FGFR2 in the patient reported below, is a protein of the kinetochore complex and is also associated with the mitotic spindle apparatus [6]. It is noteworthy that NDC80 mutations have been described in CCA, suggesting a role in cholangiocarcinogenesis [7]. Thus, the alteration presented here has the potential to be active on several different layers and warrants a thorough clinical evaluation.

## 2. Materials and Methods

The patient's tissue was processed according to internal standards for the pathological diagnostic routine. Available tissue was the hepatic primary tumor's needle biopsy from March 2018 and a surgically dissected supraclavicular lymph node metastasis from April 2018.

### 2.1. Immunohistochemistry

For immunohistochemical analyses, 2-µm-thick histological sections were cut from formalin-fixed, paraffin-embedded tissue blocks. Sections were placed in an incubator for 30 min at 70 °C. Slides were then deparaffinized through a series of xylene and gradient alcohols to water. In case of required antigen retrieval, slides were cooked for 8 min at 110 °C in either citrate pH 6.0 or Tris-EDTA pH 9.0 solution and then placed in iced water. Afterward, slides were incubated in 1× Dako Peroxidase-Blocking Solution® (S2023) for 10 min. The primary antibody was applied in Dako Antibody Diluent® (S2022) and incubated in a humidity chamber at room temperature overnight (see Table 1 for list of applied primary antibodies). After washing with Dako washing solution® (S3006), the secondary antibody Histofine Simple Stain MAX PO® anti-rabbit or anti-goat was administered for 60 min at room temperature. After two additional washes in Dako washing solution® (S3006), the chromogenic reaction was carried out with the Dako Liquid DAB + Substrate Chromogen System® according to the manufacturer's instructions. Slides were stained in Mayer's hemalum for 10 s. Coverslips were automatically applied with

the Ventana BenchMark Ultra® (Roche, Penzberg, Germany). CK7, CK19, HepPar1, and Arginase 1 were carried out as automated stainings according to the institution's diagnostic standards using Ventana BenchMark Ultra® (Roche, Penzberg, Germany).

**Table 1.** Antibodies used in the immunohistochemical study.

| Antibody | Clone | Source | Dilution | Antigen Retrieval |
|----------|-------|--------|----------|-------------------|
| FGFR2 | SP273 | Abcam | 1:1000 | citrate |
| NDC80/HEC | polyclonal | Abcam | 1:1000 | citrate |
| pERK1/2 | D13.14.4E | Cell Signaling | 1:400 | citrate |
| p4E-BP1 | 236B4 | Cell Signaling | 1:50 | citrate |
| pFRS2 | polyclonal | Abcam | 1:400 | citrate |
| pSTAT3 | D3A7 | Cell Signaling | 1:200 | Tris-EDTA |
| pPLCγ | D25A9 | Cell Signaling | 1:100 | citrate |
| CK7 | OV-TL12/30 | Dako | 1:400 | - |
| CK19 | KS19.1 | Progen | 1:400 | - |
| HepPar1 | OCH1E5 | Dako | 1:500 | - |
| Arginase1 | 380R-15 | Cell Mark | 1:50 | - |

*2.2. Image Acquisition*

The slide scanner Pannoramic 250 Flash III® (Sysmex) was used for image acquisition. The 20× magnification objective was selected. Stitched images were visualized using the CaseViewer® (Sysmex, Kobe, Japan) software. Screenshots of relevant regions were generated with a 300 ppi resolution.

*2.3. Fluorescence In Situ Hybridization (FISH)*

Paraffin-embedded, formalin-fixed tissue sections were placed for 25 min at 72 °C. Slides were then deparaffinized through a series of xylene and gradient alcohols to water. A pepsin solution was applied for 5 min at 37 °C. ZytoLight SPEC FGFR2 Dual Color Break® (ZytoVision, Bremerhaven, Germany)probe was administered. The slide was placed on a heating plate at 73 °C for 10 min for denaturation. Hybridization was performed in a humidity chamber at 37 °C overnight. 2× saline-sodium citrate buffer was used at 73 °C in a water bath the next day. After drying, a 4′,6-diamidino-2-phenylindole stain was carried out. Fluorescence images were captured on an IX73® fluorescent microscope (Olympus, Shinjuku, Tokio, Japan) with excitation and emission at appropriate wavelengths.

*2.4. Next-Generation DNA and RNA Sequencing*

Nucleic acids were isolated in an automated process using the Maxwell® RSC RNA FFPE Kit and MaxWell® RSC device (Promega, Fitchburg, WI, USA). The concentration of RNA was measured using the Agilent TapeStation® (Agilent Technologies, Santa Clara, CA, USA) and a Qubit® fluorometer (ThermoFisher Scientific, Waltham, MA, USA). For multigene-panel next-generation RNA sequencing, a PCR-based library was generated using the Archer FusionPlex Lung Kit® (Illumina, San Diego, CA, USA). The MiSeq® sequencer (Illumina, San Diego, CA, USA) was employed.

For multigene-panel next-generation DNA sequencing, a targeted multiplex PCR-based DNA enrichment was performed using the Human Actionable Solid Tumor Panel Kit® (Qiagen, Düsseldorf, Germany). The MiSeq® sequencer (Illumina, San Diego, CA, USA) was used for sequencing.

## 3. Case Report

A 63-year-old female patient was diagnosed with systemically disseminated poorly differentiated (G3) intrahepatic CCA in March 2018. The initial Union for International Cancer Control (UICC) stage was IVb, the initial TNM classification cT1b, cN1, cM1 [8]. Radiologically, metastases had been detected initially in para-aortic and left supraclavicular lymph nodes and in the left lower lobe of the lung, with the primary tumor being situated

in the right liver lobe by magnetic resonance imaging and positron emission tomography in combination with computed tomography (PET/CT) with a histological sampling of a surgically dissected supraclavicular lymph node and a liver biopsy. The tumor strongly expressed cytokeratin 7 and 19, and displayed only a weak and partial expression of cytokeratin 20, while being negative for GATA3 and TTF1 (Figure 1a–e). Furthermore, the tumor was entirely negative for Arginase1 and HepPar1, which excluded the possibility of a mixed hepatocellular cholangiocarcinoma. The patient underwent palliative systemic chemotherapy with 9 cycles of gemcitabine + cisplatin, leading to intermittent tumor regression. Increasing hematologic toxicity led to a dose reduction, and chemotherapy was continued for another 4 cycles until progressive disease in January 2019. Second-line therapy with 7 cycles of fluorouracil, folinic acid, and irinotecan (FOLFIRI) was administered until progression in March 2019. The patient received a trans-arterial chemoembolization (TACE) procedure in June 2019. These treatments led to another disease stabilization, as determined by computed tomography. Since locally ablative therapy does not treat metastatic lesions, systemic chemotherapy was resumed in August 2019. A total of 8 cycles of 75% Docetaxel were administered. After the sixth cycle, the disease progressed again. While still being treated with Docetaxel, the patient proved eligible for our molecular tumor board, and molecular analyses were initiated (Figure 1f).

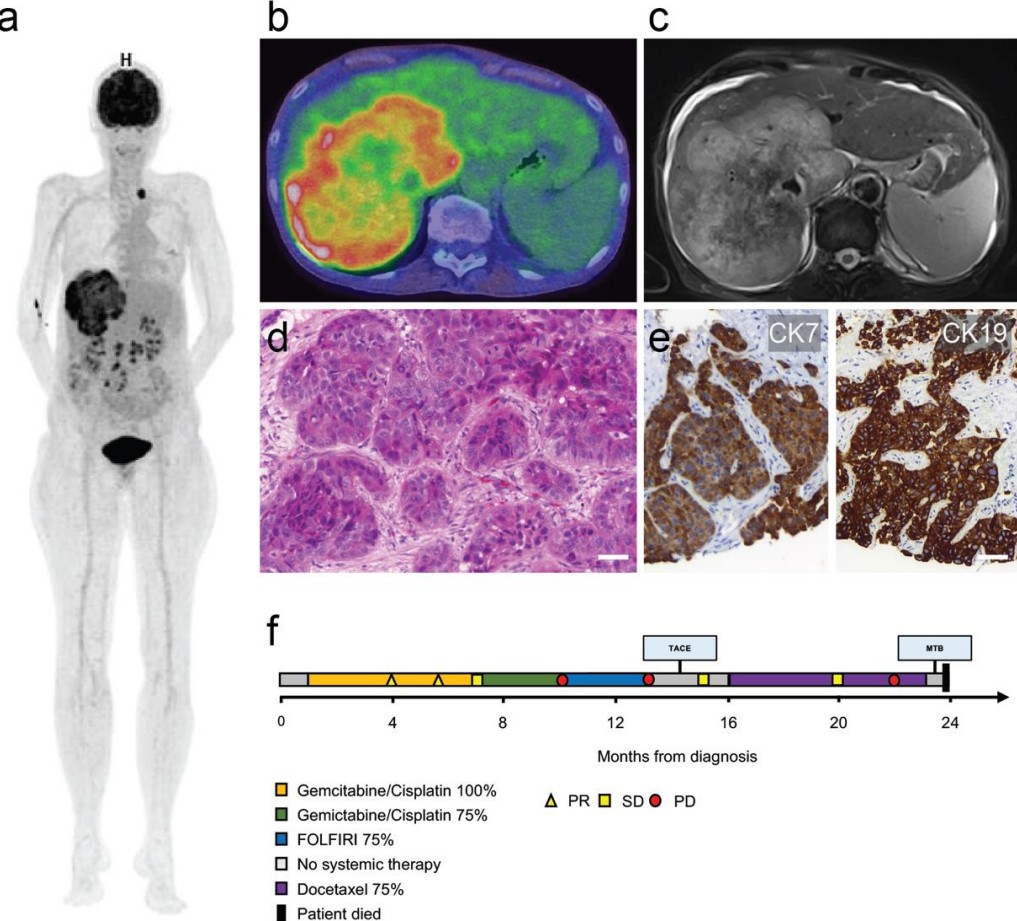

**Figure 1.** Imaging, histology and clinical course. (**a**) Three-dimensional positron emission tomography–computed tomography reconstruction showing the large primary tumor mass in the right liver lobe and a left supraclavicular nodal metastasis. (**b**) PET/CT overlay axial section of primary liver tumor. (**c**) Corresponding axial T2-weighted magnetic resonance imaging section. (**d**) Histological section of the nodal metastasis with infiltrates of poorly differentiated cholangiocarcinoma. (**e**) Cytokeratin 7 (CK7/left panel) and cytokeratin 19 (CK19/right panel) immunohistochemistry of primary tumor. Scale bars = 50 μm. (**f**) Swimmer plot illustrating clinical course (PR = partial response, SD = stable disease, PD = progressive disease).

Multigene panel next-generation DNA sequencing with the Human Actionable Solid Tumor Panel (Qiagen) did not yield any pathogenic mutation. In contrast, the multigene panel next-generation RNA sequencing using FusionPlex Lung Kit (Archer) uncovered an in-frame fusion product FGFR2-NDC80. The breakpoint was located at FGFR2 exon 17 (Homo sapiens fibroblast growth factor receptor 2, transcript variant 1, NCBI Reference Sequence: NM_000141.4) and NDC80 exon 13 (NDC80 kinetochore complex component, NCBI Reference Sequence: NM_006101.2). Consequently, the inferred fusion protein retains most FGFR2, including its intracellular kinase domains [9] with an attached C-terminal fragment of NDC80, which is implicated in kinetochore-microtubule binding [10]. The fluorescence in situ hybridization (FISH) analysis demonstrated next to normal break-apart signals (1F1R1G) and many break-apart signals with an atypical pattern (0F2R2G, 0F2R1G, or 0F1R2G). To further characterize this fusion, immunohistochemical analyses with an FGFR2 antibody and an NDC80 antibody were carried out. We detected a pronounced membranous expression of FGFR2 and an ectopic expression of NDC80 at the cell membrane, which supported the translation and the integrity of the predicted fusion protein (Figure 2). These immunohistochemical staining patterns were compared to surrounding liver tissue and a tissue microarray of CCAs, which showed generally weaker staining for FGFR2 and did not display membranous NDC80-positivity.

Immunohistochemical staining of additional FGFR2 downstream proteins was conducted. The phosphorylated effector protein FRS2 displayed intense membranous staining (Figure 3a), which was unique when compared to CCA samples on a tissue microarray ($n$ = 24) (Figure 3b). Furthermore, phosphorylated extracellular-signal-regulated kinase 1 and 2 (ERK1/2) (Figure 3c, first panel), phosphorylated PLC (Figure 3c, second panel) as well as the mTOR complex 1 (mTORC1) effector phosphorylated eukaryotic translation initiation factor 4E-binding protein 1 (4E-BP1) (Figure 3c, third panel) were all concomitantly overexpressed when compared to equally stained normal liver tissue and the cholangiocarcinoma CCA. In comparison, immunohistochemistry displayed only mild phosphorylation of signal transducer and activator of transcription 3 (STAT3) (Figure 3c, fourth panel).

Based on these conclusive results of FGFR2-NDC80 fusion protein expression and downstream effector activation (Figure 3d), we recommended using Pemigatinib in our molecular tumor board, referring to preliminary results of the FIGHT-202 study [4]. Unfortunately, the patient passed away due to biliary cancer in March 2020 before the drug could be obtained legally in Germany.

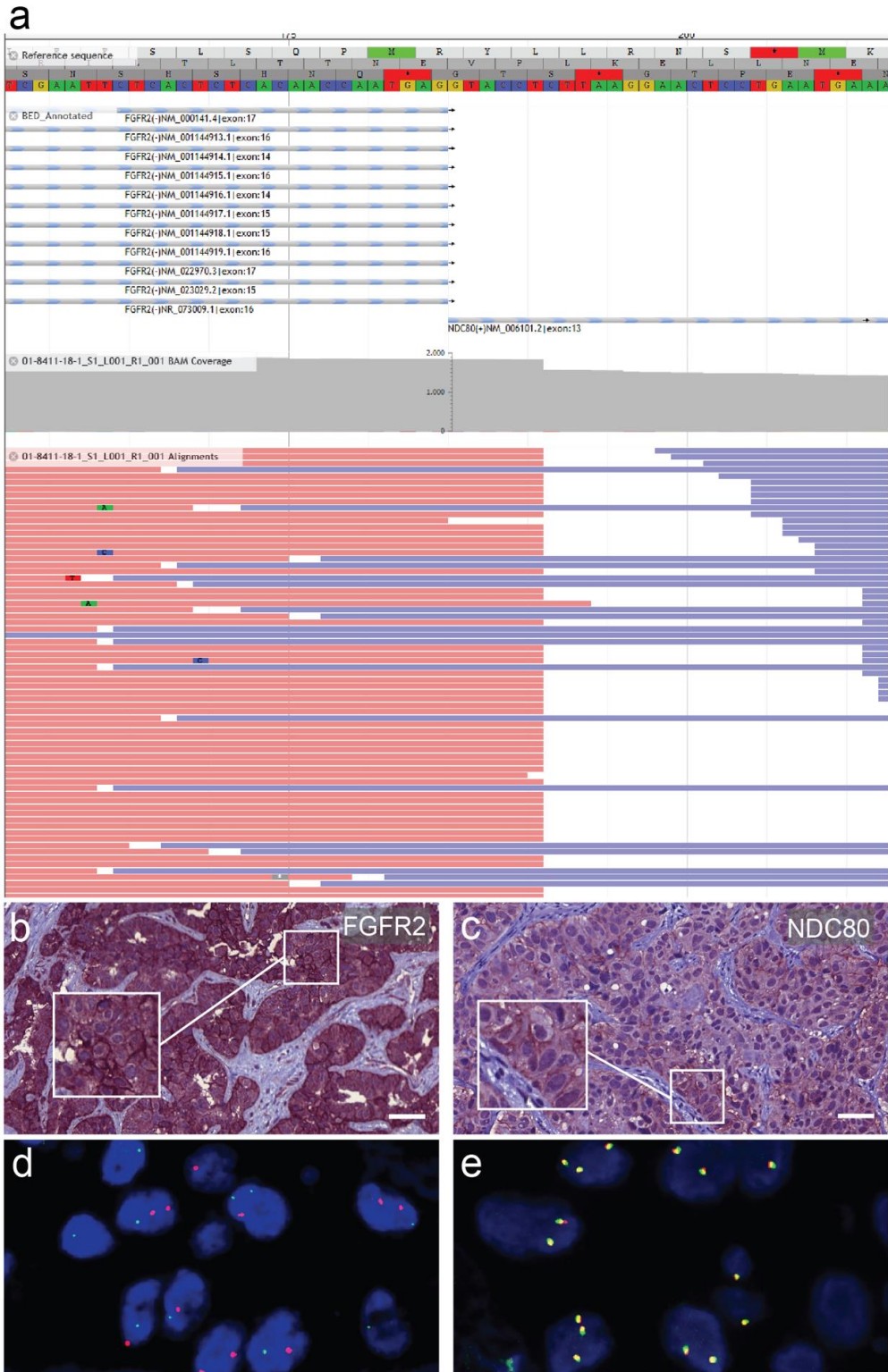

**Figure 2.** Detection and characterization of the fusion protein. (**a**) Schematic representation of the Fibroblast growth factor receptor 2 (FGFR2)-NDC80 kinetochore complex component (NDC80) fusion of the patient displaying reads and covered sequence. (**b**) Immunostaining for FGFR2 of a primary liver tumor biopsy specimen showing pronounced membranous expression. (**c**) Immunostaining for NDC80 of the nodal metastasis with focally accentuated ectopic membranous expression pattern. Scale bars = 50 μm. (**d**) FGFR2 fluorescence in situ hybridization break-apart signal with atypical distribution pattern in lymph node metastasis of the presented patient. (**e**) Physiologically fused signals in a different cholangiocarcinoma patient for comparison.

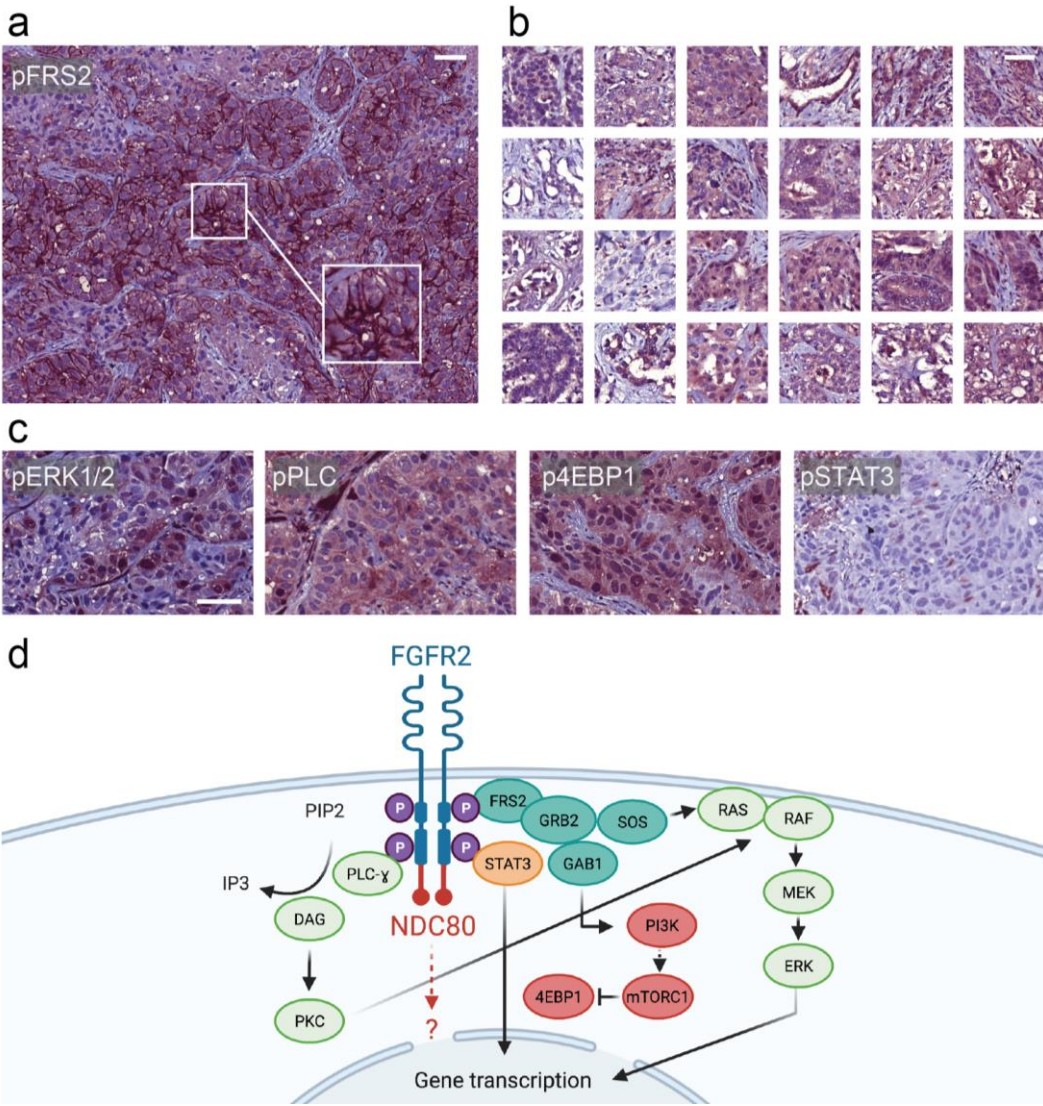

**Figure 3.** Evaluation of downstream effector proteins. (**a**) pFRS2 immunostaining of the nodal metastasis showing strong membranous positivity. (**b**) pFRS2 immunohistochemistry of a self-made tissue microarray of cholangiocarcinoma specimens lacking strong membranous staining. (**c**) From left to right: pERK1/2-, pPLC-, p4EBP1-, and pSTAT3-immunohistochemistry of the nodal metastasis. Scale bars = 50 μm. (**d**) Schematic representation of a hypothetical ligand-independent activation of the FGFR-NDC80 fusion with downstream effectors.

## 4. Discussion

Rare fusion proteins frequently occur in molecular tumor boards and pose a significant challenge to clinicians evaluating their pathogenicity and therapeutic potential. Here, we present an exemplary case of an FGFR2-NDC80 fusion. Notably, more than half of the discovered FGFR fusions are yet unreported [11], and the question of whether these represent mere bystanders or actual oncogenic drivers is open to debate. A thorough analysis of the discovered fusion was performed to evaluate its oncogenic significance and therapeutic actionability in the current case. Several levels of evidence support the oncogenic potential of the FGFR2-NDC80 fusion. First, on a structural level, the sequence retained the kinase domain of FGFR2 as it is known for other FGFR fusion proteins [5], which should allow intact downstream signaling.

Moreover, the mechanism of mistargeting FGFR fusion proteins to the spindle apparatus has been reported previously for members of the transforming acidic coiled-coil-containing protein (TACC) family, leading to chromosomal instability [12]. A simi-

lar mechanism could be envisaged for NDC80, which is a kinetochore component with microtubule-binding domains. Interestingly, we also demonstrated a FISH break apart with a partly aberrant signal distribution, that could be explained with chromosomal instability. Secondly, the fusion protein led to a relocation of NDC80-immunohistochemistry staining to the plasma membrane. At the same time, FGFR2 immunohistochemistry was also pronounced at the membrane, which is evidence of the expression of an intact fusion protein. Thirdly, downstream effectors such as pFRS2, pPLC, and p4EB-P1 displayed significantly elevated levels when compared to normal liver tissue and tissue microarrays of cholangiocarcinoma specimens. Concerning pFRS2, a substantial increase of membranous positivity was only observed in the described patient's tumor. Such a pFRS2 staining pattern has also been described before in a case report in a patient harboring an FGFR2-TACC3 fusion [13]. Overall, these findings were sufficient to recommend the administration of the specific FGFR2 inhibitor Pemigatinib. Unfortunately, the drug was still unavailable, and the patient passed away before receiving the FGFR inhibitor therapy.

This case underlines that comprehensive molecular testing in patients with aggressive malignancies like biliary tract cancer needs to be timed carefully. The window of opportunity for administering a stratified molecular therapy closes quickly after progression on "last line" therapy in these patients, and applications for these usually costly drugs can easily take up to three months. In our institution, we, therefore, initiate molecular testing when starting the "last line" therapy to conduct a thorough analysis and give clinicians the time to obtain any recommended (experimental) medication. Although the described method of verifying fusions with immunostainings and correlating these to microarrays appears cost and time-intensive, we still believe that specialized centers offering molecular tumor boards for tailored therapies could often meet the technical requirements for such analyses, especially in the case of a subspecialization in certain cancer entities. What should encourage us is scientific vigor and the promising outcomes that can be attained by pharmacologically targeting fusion proteins.

## 5. Conclusions

In this case report, we described a CCA harboring a novel FGFR2-NDC80 fusion. To verify the oncogenic potential of the observed fusion protein, immunohistochemical analyses of the fusion partners and the downstream effectors were performed, which showed an aberrant expression of the fused proteins in combination with an upregulation of the downstream effectors. To evaluate these findings qualitatively, immunohistochemical patterns were compared to a tissue microarray of cholangiocarcinoma specimens. With this case report, we hope to offer clinicians and pathologists some guidance in evaluating rare fusions and their clinical actionability.

**Author Contributions:** Conceptualization, A.S., K.U., M.E., T.P. and D.F.C. Methodology, A.S., K.U. and D.F.C., investigation, A.S., F.K., F.L., K.U., W.D., A.K., S.S.; resources, J.G., N.V., S.O., W.D.; data curation, A.S., W.D.; writing—original draft preparation, A.S.; writing—review and editing, all authors; visualization, A.S., F.L., J.G.; supervision, K.U., D.F.C., M.E. All authors have read and agreed to the published version of the manuscript.

**Funding:** This research received no external funding.

**Institutional Review Board Statement:** The study was conducted according to the guidelines of the Declaration of Helsinki and approved by the Ethics Committee of the University of Regensburg (Molecular Tumor Board Registry Study, protocol code 20-1682-101).

**Informed Consent Statement:** Written informed consent was obtained from the patient.

**Data Availability Statement:** Additional data is available on request.

**Acknowledgments:** The illustration in Figure 3d was created with BioRender.com (accessed on 7 March 2021).

**Conflicts of Interest:** The authors declare no conflict of interest.

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
