# Peer review of "Identification and In-Depth Analysis of the Novel FGFR2-NDC80 Fusion in a Cholangiocarcinoma Patient: Implication for Therapy"

_curroncol, doi:10.3390/curroncol28020112_

Round 1

Reviewer 1 Report

In this manuscript by Scheiter et al., the authors have presented an unique FGFR2-NDC80 Fusion in a Cholangiocarcinoma patient. They observed that the FGFR2-NDC80 fusion leads to strong activation of the
FGFR2 signaling pathway. In addition, the expression of downstream effectors was also increased. This is an interesting observation. However, the following points should be addressed before reconsidering the manuscript for publication.

  1. What is the rationale to select NDC80 for the fusion study?
  2. The membranous expression of FGFR2 and NDC80 were not clearly visible in figure 2. The authors should provide high magnification images.
  3. Clear images of downstream effectors should be provided in figure 3 to show the localization. 
  4. FGFR genetic aberrations should be confirmed using Fluorescent in-situ hybridization (FISH).

Reviewer 2 Report

The work by Scheiter et al. constitutes a case report in which the authors identify a new FGFR2-NDC80 fusion in a patient with intrahepatic cholangiocarcinoma (iCCA). The patient was diagnosed with advanced and metastasized iCCA, which was shown to harbor FGFR2-NDC80 fusion after several cycles of chemotherapy with GemCis. In fact, the authors herein described that this fusion results in the overactivation of the FGFR2 pathway and recommended to patient to be treated with pemigatinib, a FGFR2 inhibitor, although the patient passed away before receiving the therapy. The manuscript is well-written and the analysis were nicely conducted. Still, there are several points that should be addressed in order to increase the overall quality of this case report.     

  • Did the authors evaluated the possibility of this tumor being a mixed HCC-iCCA tumor? Was this diagnosis discarded?
  • In the IHC images, if possible, it would be of great value to also display the stainings in the surrounding tissue, to ascertain if the FGFR2 pathway is actually increased.
  • Is this fusion somehow different from the already described ones? In terms of biological effect, this fusion is different form the other ones?
  • A methods section should be included.
  • In the last sentence of results, should it be March 2020?

Round 2

Reviewer 1 Report

The authors have addressed all the concerns and the manuscript was improved substantially. The manuscript can be accepted for publication.